# FILNet: Fast Image-Based Indoor Localization Using an Anchor Control Network

**DOI:** 10.3390/s23198140

**Published:** 2023-09-28

**Authors:** Sikang Liu, Zhao Huang, Jiafeng Li, Anna Li, Xingru Huang

**Affiliations:** 1School of Communication Engineering, Hangzhou Dianzi University, Hangzhou 310018, China; sikangliu@whu.edu.cn; 2State Key Laboratory of Information Engineering in Surveying, Mapping, and Remote Sensing, Wuhan University, Wuhan 430072, China; jeff_whu@126.com; 3School of Electronic Engineering, Queen Mary, University of London, London E1 4NS, UK; zhao.huang@qmul.ac.uk (Z.H.); anna.li@qmul.ac.uk (A.L.)

**Keywords:** indoor localization, anchor control network, affine invariance enhancement, feature matching, fast spatial indexing

## Abstract

This paper designs a fast image-based indoor localization method based on an anchor control network (FILNet) to improve localization accuracy and shorten the duration of feature matching. Particularly, two stages are developed for the proposed algorithm. The offline stage is to construct an anchor feature fingerprint database based on the concept of an anchor control network. This introduces detailed surveys to infer anchor features according to the information of control anchors using the visual–inertial odometry (VIO) based on Google ARcore. In addition, an affine invariance enhancement algorithm based on feature multi-angle screening and supplementation is developed to solve the image perspective transformation problem and complete the feature fingerprint database construction. In the online stage, a fast spatial indexing approach is adopted to improve the feature matching speed by searching for active anchors and matching only anchor features around the active anchors. Further, to improve the correct matching rate, a homography matrix filter model is used to verify the correctness of feature matching, and the correct matching points are selected. Extensive experiments in real-world scenarios are performed to evaluate the proposed FILNet. The experimental results show that in terms of affine invariance, compared with the initial local features, FILNet significantly improves the recall of feature matching from 26% to 57% when the angular deviation is less than 60 degrees. In the image feature matching stage, compared with the initial K-D tree algorithm, FILNet significantly improves the efficiency of feature matching, and the average time of the test image dataset is reduced from 30.3 ms to 12.7 ms. In terms of localization accuracy, compared with the benchmark method based on image localization, FILNet significantly improves the localization accuracy, and the percentage of images with a localization error of less than 0.1m increases from 31.61% to 55.89%.

## 1. Introduction

Currently, location-based services (LBS) based on indoor positioning show significant potential in large indoor environments such as airports, underground parking lots, shopping malls, and hospitals [1]. For instance, in the context of building management and operation, Bluetooth low-energy-based indoor positioning services can provide precise occupancy information during emergency management, facilitating effective response and rescue operations [2]. Furthermore, through sensor data and machine learning algorithms, personnel prediction and location-based services can enhance safety and comfort [3]. Simultaneously, in the realm of energy efficiency within buildings, indoor positioning services not only enable the automatic management of users’ socket loads [4] but also allow the adjustment of heating, ventilation, and air conditioning systems based on devices connected through WiFi [5]. Lastly, location services assist researchers in obtaining Point of Interest (POI) data from various sources and integrating them for analysis and comparison [6]. Therefore, indoor positioning systems have found widespread applications across various research domains. 

With the rapid development of pervasive and mobile computing, a series of signals for indoor localization have been explored, such as Bluetooth-based signals [7,8], WiFi signals [9,10], and geomagnetic signals [11]. Meanwhile, different communication protocols and technologies can work with each other in indoor localization field. ZigBee, as a low-power and low-data-rate wireless communication protocol, is important for indoor localization. Firstly, the ZigBee protocol enables bi-directional communication, which allows reliable communication between sensors and control systems. Secondly, the ZigBee protocol also has a secure network connection, which can ensure the security of data transmission. Then, the ZigBee protocol can perform distance estimation using the receiving signal strength indicator (RSSI) for indoor positioning. Finally, ZigBee can be combined with WiFi, Bluetooth, geomagnetism, and images that work together to improve the accuracy and reliability of indoor positioning systems.

Bluetooth-based indoor localization methods have low power consumption and can run for a long time on mobile devices, such as IoT devices or portable devices. It is also universal, for example, smartphones support Bluetooth technology for localization services. However, Bluetooth-based localization methods require the deployment of a large number of Bluetooth beacon devices in indoor environments due to high signal attenuation and limited transmission range [12], which can increase the cost of deployment and maintenance. In addition, WiFi-based signal localization methods have high accuracy in small areas. They achieve high-precision localization by constructing and maintaining a library of signal fingerprints, which is useful for applications such as those in hospitals that require small-scene indoor navigation and precise location information. However, fingerprint WiFi-based methods [13,14] are susceptible to environmental factors, and signal fluctuations lead to unstable signal strength data in the fingerprint library, which can severely degrade the performance of direct signal strength (RSS) matching (RDM) [15]. Geometric WiFi-based methods use geometric methods that are susceptible to non-line-of-sight (NLOS) and multipath effects in measuring and estimating geometric features, which reduces localization accuracy. Then, geomagnetic-based localization methods offer low power consumption, while RF signals penetrate buildings and are not easily affected by walls or other obstacles. However, geomagnetism can only provide limited accuracy, typically in the range of a few meters to a dozen meters. This makes it potentially less accurate in applications that require high-precision localization.

Unlike the above methods, image-based localization methods not only provide high localization accuracy but also take into account low power consumption, which is more suitable for mass users completing localization using smartphones.

However, the usual image-based localization methods are limited with respect to privacy issues, especially in establishing the fingerprint database. Therefore, this paper uses an anchor image database to replace the image database of scene collection. The content of the anchor image can be set by itself according to the complexity of the scene environment, which effectively avoids the limitations of scene privacy as well as personal privacy. In addition, the fingerprint database of this paper stores the local features of anchor images, and the selected SURF local features are robust to rotation, scale change, and illumination change. Meanwhile, the proposed affine invariant enhancement algorithm based on multi-angle screening and supplementation of features is applied on the basis of SURF, which solves the problem of image perspective transformation and improves localization accuracy. Therefore, good localization results can be accomplished as long as there exists a number of matching feature pairs that satisfy the conditions in case of line-of-sight changes or partial occlusion. Finally, to address the low efficiency of existing image-based localization methods, this paper proposes a fast spatial indexing algorithm to speed up the feature matching process. Meanwhile, the method in this paper does not require expensive hardware support and only requires a camera-equipped cell phone and the collection of selected anchor images to accomplish high-precision and high-efficiency indoor localization.

Image-based indoor localization methods mainly include two stages: the offline feature database construction stage and online location query [16]. In the offline stage, visual features are extracted (e.g., Speeded-Up Robust Features (SURF) [17]) to construct a feature fingerprint database. In the online stage, the features from the input image are matched with the previously constructed database, and the location of the input image is obtained. As for feature database construction, many previous studies [18] did not consider the affine invariance problems, and multi-view features were not complemented. However, when performing the matching between the input image and the feature library, all features in the feature library are directly matched, which has low efficiency and high time consumption. In addition, no feature screening is performed after matching, which may cause many errors in matching.

To address the aforementioned problems, this paper proposes a fast image-based indoor localization method based on an anchor control network named the FILNet, which consists of two stages, the offline stage and the online stage, as shown in Figure 1. In the offline stage, first, murals are used as anchors and are deployed in an indoor environment at a suitable distance. Next, some of them are selected as control anchors to establish the anchor control network, and these control anchors are measured by the total station to record the coordinates of their four corner points. Further, a visual–inertial odometer (VIO) based on the ARcore is used to infer the corresponding coordinate information of the rest of the anchors through detailed surveys and extract their descriptors. To enhance the affine invariance of anchors, the main-direction image screening algorithm is adopted to obtain multi-angle features for a single feature point. Finally, the feature fingerprint database of the anchors is constructed by fusing all features obtained in the previous steps. In the online stage, when a user inputs a positioning image, two-dimensional (2D) image features are extracted and matched with the three-dimensional (3D) features of anchors from the established database. Next, anchors with higher matching precision are selected as active anchors, and the fast spatial indexing of the active anchors is performed to decrease the time consumption of feature matching. Further, a homography matrix is introduced to correct the errors. Finally, the user’s position is determined.

The main contributions of this work can be summarized as follows.

(1)An efficient anchor feature fingerprint database construction scheme is designed based on an anchor control network. The proposed scheme has low requirements for equipment and environment and high matching efficiency and is concise;(2)A multi-angle features supplementary algorithm based on the main-direction image, similar to the ASIFT algorithm, is proposed. This algorithm is based on the image POS and cuts images from multiple views to enrich a single feature point and improve the recall rate of feature matching even when the perspective changes significantly;(3)A fast spatial indexing algorithm is designed to improve feature matching efficiency to decrease time consumption, and a homography matrix is introduced to verify the correctness of matching using the projection error rate and delete the error matching points.

The rest of this paper is organized as follows. Section 2 presents the related research on visual indoor localization. Section 3 describes the proposed method in detail. Section 4 presents the experiment results. Finally, Section 5 concludes the paper and presents future work directions.

## 2. Related Work

This section describes the related research on feature matching and the indoor localization approaches used to develop the proposed method.

### 2.1. Feature Matching

In recent years, feature matching algorithms have been extensively used to eliminate outliers from putative sets, and they can be roughly divided into three groups: resampling methods, non-parametric interpolation methods, and graph matching methods [19].

Ma et al. [20] presented a neighborhood manifold representation consensus (NMRC) method to match features robustly while considering the stable neighborhood topologies of true potential matches. However, the main limitation of this method is that representing the neighborhood manifold is challenging when putative correspondences are scarce locally. Our current work aims to address this limitation by introducing spatial information. An innovative point correspondence identification model based on the correspondence function (ICF) was developed by Li and Hu [21]. The ICF aims to reject mismatches from provided putative point correspondences by comparing them with the estimated correspondence functions. However, this method is vulnerable to severe degeneration in the presence of a considerable number of outliers. In our research, we use a multi-view feature complement algorithm to improve the matching rate.

An efficient vector field consensus algorithm was designed by Ma et al. [22] to establish robust point correspondences between two sets of points, which imposes non-parametric geometrical constraints on the correspondence as a prior distribution and uses Tikhonov regularizers in reproducing the kernel Hilbert space. A Gaussian field consensus (GFC) model was introduced by Wang et al. [23] to reject a robust outlier from putative point set matching correspondences. The GFC extends the Gaussian mixture distance to non-parametric models and then reduces it to a single Gaussian model. However, the non-parametric approaches have the shortcoming of high computational complexity.

Sarlin et al. [24] presented an attention-based graph neural network named the SuperGlue to match two sets of local features by concurrently identifying correspondences and rejecting unmatchable points. Jiang et al. [25] developed an improved graph learning–matching network (GLMNet) model for graph matching. The GLMNet integrates graph learning and graph matching architectures into a single end-to-end network to learn a pair of ideal matching graphs that are most effective for graph matching. However, this model is not limited to only two-graph matching but can also be applied to multi-graph matching.

### 2.2. Indoor Localization

Indoor positioning based on smartphones has become the mainstream localization method due to its convenience and popularity. These methods include the MEMS- and image-based methods.

Qian et al. [26] designed an enhanced indoor localization technique based on inertial sensors of a smartphone purchased off the market, which can achieve high precision, but the heading determination performance should be improved. Wang et al. [27] used the magnetic and light sensors of smartphones to develop a deep long short-term memory (LSTM) network-based system for indoor localization. However, this approach is limited by the change in light and building materials. To achieve accurate real-time localization, Murata et al. [28] proposed and implemented an improved probabilistic localization system using a smartphone’s inertial sensors. The main drawback of this approach is that the localization range is limited, and additional equipment is needed to assist in localization. Ashraf et al. [29] demonstrated how an ensemble deep neural network (NN) could be used in combination with heterogeneous sensors to perform magnetic-field-based indoor localization. Although this method can obtain promising results, its accuracy degrades with various devices.

Niu et al. [16] proposed a highly automated image-based localization algorithm (HAIL), which can be installed on mobile devices. The HAIL not only performs image confirmation after taking images automatically but also improves accuracy by using motion sensors and map constraints. A low-cost image-based indoor navigation system based on 3D sensor-enriched models intended for the indoor environment was presented by Dong et al. [18]. This system can solve the missing pedestrian path problem but does not consider the image view changes. Shu et al. [30] presented an image-based localization method which uses a built-in MEMS in a mobile device held by a moving pedestrian. This method can solve the problem of resource limitation and can improve the effectiveness and practicality with the assistance of smartphone MEMS. However, the more powerful processing performance on a mobile device should be further considered. A model of localizing images captured from two traversals of the same area under both day and night conditions was introduced by Anoosheh et al. [31]. This approach employs a modified image translation model to convert nighttime driving images into more useful daytime representations, but it lacks different image features and requires high-quality images. Baek et al. [32] introduced an augmented reality (AR)-based system into facility management. This approach uses an image-based indoor localization technique to estimate a user’s position and orientation by comparing the user’s perspective to building information modeling (BIM).

## 3. Proposed Model

### 3.1. The Overview of the Proposed Approach

To improve the feature matching efficiency and multi-view positioning accuracy in indoor environments, this paper uses the concept of an anchor control network to design a fast image-based indoor localization algorithm.

As shown in Figure 2, the framework of this proposed algorithm includes two main phases, the offline phase and the online phase. In the offline phase, anchors are set up in an indoor environment, and some of them are selected as control anchors. The selected anchors are measured, and their spatial properties are recorded. Meanwhile, the rest of the anchors are inferred by detailed surveys based on VIO data. Finally, the feature database is constructed using affine invariance enhancement technology. In the online phase, a user inputs a positioning image, and the input image is de-distorted first, and then image features are extracted. The extracted features are matched and screened with those in the fingerprint database, and the feature matching is performed. Then, the pose of the smartphone camera (i.e., user position) is calculated.

### 3.2. Feature Selection and Extraction

The processes of feature extraction and selection are some of the most important tasks in fingerprint database construction. Different features and feature extraction algorithms can achieve different efficiency in feature matching. Therefore, knowing how to select appropriate features and an optimal feature extraction algorithm has been a key challenge. Image features can be roughly divided into local and global features. Global features include the image description from the aspect of overall image attributes, including color, texture, and shape. Local features represent only the local properties of an image. However, the correlation between local features is low and relatively stable in complex environments, so they are more suitable for the establishment of an anchor fingerprint database.

The scale-invariant feature transform (SIFT) feature extraction algorithm has been widely used to detect local features in images [33]. This algorithm determines key points in different scale spaces and calculates the directions of key points. The key points do not change with the lighting, affine transformation, and noise conditions, which includes corner points, edge points, bright spots in dark areas, and dark spots in bright areas. Meanwhile, this algorithm has a large scale, good rotation invariance, and strong robustness against blur. However, this algorithm is time consuming and has low efficiency. In addition, feature points are represented by 128-dimensional vectors. As an improved version of the SIFT, in the SURF, feature points are represented by 64-dimensional vectors, which greatly improves the detection efficiency compared to that of the original SIFT. In addition, compared to the two above-mentioned algorithms, the oriented FAST–rotated BRIEF (ORB) algorithm [34] is much faster, but it lacks scale transformation performance. A detailed comparison of the three algorithms is summarized by [35] and is shown in Table 1. As can be seen from this table, the SURF feature extraction algorithm has the best rotation transformation, angle transformation, and scale transformation; therefore, it could be considered the best choice to improve the efficiency of feature extraction.

The specific steps of the feature extraction process are as follows:(1)Scale-space interest point detection: The SURF uses a Hessian matrix to detect feature points. The Hessian matrix is a square matrix composed of the second partial derivative of a multivariate function, which describes the local curvature of the function. Equation (1) defines the Hessian matrix of the image I(x,y), where H denotes the Hessian matrix with image feature point coordinates I(x,y).
(1)$$H(I(x,y))=[\begin{array}{l} \frac{\partial^{2}I(x,y)}{\partial x^{2}}\\ \frac{\partial^{2}I(x,y)}{\partial x\partial y} \end{array}\begin{array}{c} \frac{\partial^{2}I(x,y)}{\partial x\partial y}\\ \frac{\partial^{2}I(x,y)}{\partial y^{2}} \end{array}]$$

The determinant in Equation (1) represents the variation around a pixel point, and the extreme value of the determinant denotes a feature point. The square filter is used to replace the Gaussian filter in the SIFT, and the four corners’ values located in the square filter are calculated based on the integral graph, thus improving the operation speed significantly;

(2)Interest point position: The non-maximum suppression is used to localize interest points in an image and over scales. The interpolation algorithm proposed by Brown and Lowe [36] is employed to interpolate the maxima of the determinant of the Hessian matrix into scale and image space;(3)Interest point orientation assignment: A vector is constructed by summing the transform values in the x-and y-direction inside an angle interval of the x–y plane and computing the Hal wavelet transform of the pixels surrounding the feature point in the x- and y-direction. The direction of the feature point is the longest vector, which is the vector with the largest x and y components;(4)Feature descriptor calculation: In the descriptor extraction process, the first step is to construct a square region centered around the interest point and oriented along the direction selected in the previous section. Further, a 5 pixel × 5 pixel region is set as a sub-region, and 20 × 20 pixels around the feature points are extracted, which is a total of 16 sub-regions. Then, the sum of the Hal wavelet transforms and its vector length in the x- and y-direction is obtained. At this time, the direction of parallel feature points is the x-direction, and the direction of vertical feature points is the y-direction within the sub-region: Σdx, Σdy, Σ|dx|, and Σ|dy|, which can generate a 64-dimensional descriptor.

### 3.3. Anchor Fingerprint Database Construction

The anchor fingerprint database construction is a key task in the proposed indoor localization system. This section describes the construction process of a robust feature fingerprint database in detail, which includes control anchor measurement, anchor detailed surveys, and feature multi-view supplementation.

#### 3.3.1. Control Anchor Measurement

In this study, murals are used as anchors which are mainly distributed in various areas of the interior, such as corridors, rooms, and aisles. When the anchors are deployed, accurate information on the object coordinates of the four corners of each anchor is needed to establish a feature database. However, if anchors are measured individually, that is time consuming and laborious, and thus will not be able to meet the efficiency requirement of the database construction in large-scale scenes. Therefore, this paper follows the idea of a measurement control network, takes anchors as basic research objects, and proposes the concept of the anchor control network. The control anchors are measured first, and then the VIO is used to infer the four corner coordinates of the rest of the anchors according to the information on the control anchors.

To record the position information of the anchors, the object coordinates of the four corner points (P1, P2, P3, and P4) of the rectangle murals are measured by the total station, but some murals are non-rectangular. Therefore, to simplify the measurement process, the coordinates of the four corners of the outer rectangle are also measured. Then, their SURF features are extracted. The measurement process is illustrated in Figure 3.

#### 3.3.2. Anchor Detailed Survey

The traditional detailed survey schemes have been mainly based on the polar coordinate method, direction intersection method, and distance method. Compared to the traditional methods, the detailed surveys based on an anchor control network use the visual–inertial odometer (ARcore) combined with the spatial 3D similarity transformation to measure coordinate information and features. However, the real-time image, position, and attitude of a smartphone can be obtained through the VIO based on the ARcore platform. The reference system of pose information is based on a smartphone’s coordinate system, so it is also necessary to use the control anchor to perform a similar 3D transformation to convert the coordinates of the detailed surveys to the target coordinate system. A flowchart of the detailed surveys is shown in Figure 4.

First, the path of the detailed surveys is planned according to the distribution of control anchors and location information. When traveling along the route planned with ARcore, images from the control anchors and other types of anchors are intercepted one after another, and their pose information is recorded. Second, the four corner pixel coordinates of the control anchors and the remaining anchors are extracted from the input image. Then, their object coordinates in the smartphone coordinate system are calculated by the forward intersection algorithm based on the image pose information. To obtain more accurate coordinates, a certain distance should be maintained between images of the forward intersection. The calculation process can be expressed as follows:(2)$$\left[\begin{array}{c} u\\ v\\ 1 \end{array}\right]=\frac{1}{Z_{C}}\left[\begin{array}{cccc} f_{x} & 0 & u_{0} & 0\\ 0 & f_{y} & v_{0} & 0\\ 0 & 0 & 1 & 0 \end{array}\right]\left[\begin{array}{cc} R & t\\ 0 & 1 \end{array}\right]\left[\begin{array}{c} X_{W}\\ Y_{W}\\ Z_{W}\\ 1 \end{array}\right]$$
where (u, v) describes the pixel coordinate, R indicates the rotation matrix, t presents the translation matrix, K is the camera’s internal parameter, fx and fy denote the focal length, (Xw, Yw, Zw) denotes the object square coordinates, and Zc denotes the normalization parameter. Suppose matrix P is denoted by:(3)P=K[R|t]

Then, the relationship between the object point and the image point can be expressed by the P matrix as follows:(4)$$s\left[\begin{array}{c} u\\ v\\ 1 \end{array}\right]=P\left[\begin{array}{c} X\\ Y\\ Z\\ 1 \end{array}\right]=\left[\begin{array}{cccc} p_{11} & p_{12} & p_{13} & p_{14}\\ p_{21} & p_{22} & p_{23} & p_{24}\\ p_{31} & p_{32} & p_{33} & p_{34} \end{array}\right]\left[\begin{array}{c} X\\ Y\\ Z\\ 1 \end{array}\right]$$
where S denotes the default parameter. Therefore, the object coordinates can be obtained by solving according to Equations (2)–(5).
(5)$$\begin{array}{l} u_{1}=\frac{p_{11}X+p_{12}Y+p_{13}X+p_{14}}{p_{31}X+p_{32}Y+p_{33}X+p_{34}}\\ v_{1}=\frac{p_{21}X+p_{22}Y+p_{23}X+p_{24}}{p_{31}X+p_{32}Y+p_{33}X+p_{34}}\\ u_{2}=\frac{{p_{11}}^{\prime }X+{p_{12}}^{\prime }Y+{p_{13}}^{\prime }X+{p_{14}}^{\prime }}{{p_{31}}^{\prime }X+{p_{32}}^{\prime }Y+{p_{33}}^{\prime }X+{p_{34}}^{\prime }}\\ v_{2}=\frac{{p_{21}}^{\prime }X+{p_{22}}^{\prime }Y+{p_{23}}^{\prime }X+p_{24}}{{p_{31}}^{\prime }X+{p_{32}}^{\prime }Y+{p_{33}}^{\prime }X+p_{34}} \end{array}$$

Third, to realize the 3D similarity transformation between the geodetic coordinates and the smartphone coordinates, seven transformation parameters between the two coordinate systems are defined, including the rotation of the three coordinate axes denoted by σ,φ,ϕ, three translations denoted by ∆x, ∆y,∆z, and scale factor λ. Therefore, the anchor coordinates can be converted from the smartphone coordinates system to the geodetic coordinate system as follows:

Assume that the centroids of the two groups of points are $p_{i}$ and $p_{i}^{\prime }$, and they are respectively expressed by Equation (6).
(6)$$p_{}=\frac{1}{n}\sum_{i=1}^{n}\left(p_{i}\right),p^{\prime }=\frac{1}{n}\sum_{i=1}^{n}\left({p_{i}}^{\prime }\right)$$

Then, the de-centroid coordinates of the two groups of points can be expressed as follows:(7)$$q_{i}=p_{i}-p,{q_{i}}^{\prime }={p_{i}}^{\prime }-p^{\prime }$$

Further, define the error e by:(8)$$e_{i}=q_{i}-SR{q_{i}}^{\prime }$$

Therefore, the minimum error can be expressed by:(9)$$\begin{array}{l} \underset{R,t}{\min}\;e=\sum_{i=1}^{n}{\left\|\left(q_{i}-SR{q_{i}}^{\prime }\right)\right\|}^{2}\\ =\sum_{i=1}^{n}q_{i}^{T}q_{i}+S^{2}{{q_{i}}^{\prime }}^{T}R^{T}R{q_{i}}^{\prime }-2Sq_{i}{}^{T}R{q_{i}}^{\prime } \end{array}$$

After the SVD decomposition to 2${Sq_{i}}^{T}R{q_{i}}^{\prime }$, S can be expressed by:
(10)$$S=\sqrt[{3}]{\det(q_{i}q_{i}\prime^{T}{\left({q_{i}}^{\prime }{{q_{i}}^{\prime }}^{T}\right)}^{-1}R)}$$

According to the calculated rotation matrix R and scale coefficient S, the translation matrix t can be calculated as follows:(11)$$t=p-SRp^{\prime }$$

Since the accuracy of the ARcore can decline with the running time, the coordinate system conversion parameters obtained from the initial data may not be suitable for ending data. Therefore, the conversion parameters should be calculated using the segmented calculation scheme. Firstly, we choose some anchors as active anchors, and these active anchors are measured by total station to obtain the corresponding accurate position information. Then, we use this accurate position information to revise the cumulative errors caused by ARscore. More specifically, every piece of active anchor position information measured by total station is used as a new starting point to infer the position of other visual anchors rather than using the information position with errors inferred by ARscore due to its long runtime. Hence, the cumulative errors are narrowed, and the accuracy is improved. Meanwhile, the indoor existing control points are also used to revise the cumulative errors from ARscore.

#### 3.3.3. Feature Multi-View Affine Transformation

The SURF method is used in this study due to its advantages of rotation, scale, and brightness invariance. However, the SURF method is unstable when the view angle changes. To solve this problem, Morel et al. [37] proposed the ASIFT (AffineSIFT) algorithm, which obtains a series of simulated images through the affine simulation transformation and then performs feature detection and matching. Motivated by this algorithm, this study intercepts the relevant image from a video stream based on the image POS to realize multi-angle positioning. The main steps of this process are as follows:(1)The corresponding path of a video stream is planned according to the location distribution that may appear when a user takes images. Different from the detailed survey planning path, the goal of this process is to obtain the frontal image of anchors and then infer the coordinates of the four corners of the circumscribed rectangle of the detailed anchors. The main goal of the affine simulation planning path is to obtain images from various perspectives to complement the features of the corresponding anchor;(2)A video is obtained along the planned path, and the POS information of an image is recorded at the same time.

As shown in Figure 5, a sample collection process is performed. The peripheral arc represents the path when collecting images of different view angles around the same murals; the internal blue solid line is the anchor mural on the wall; the small blue rectangle denotes the image captured by the video stream at a certain time interval when the user walks; the dotted line represents the normal vector of the image;

(3)The image of each anchor is filtered according to the POS. The POS information obtained by the ARcore is recorded in the form of a quaternion q (w, x, y, z), and the corresponding rotation matrix R is defined by:


(12)
$$R=\left[\begin{array}{ccc} 1-2y^{2}-2z^{2} & 2xy+2wz & 2xz-2wy\\ 2xy-2wz & 1-2x^{2}-2z^{2} & 2yz+2wx\\ 2xz+2wy & 2yz-2wx & 1-2x^{2}-2y^{2} \end{array}\right]$$


The direction of the main optical axis can be regarded as the image direction, and in the camera coordinate system, the main optical axis vector α can be expressed as follows:(13)$$\alpha ={\left[0,0,1\right]}^{T}$$

Further, the 3D pose of a camera in space β can be expressed by:(14)β=R∗α

However, image screening uses only the horizontal heading angle, so only the project on a two-dimensional plane is needed to calculate the horizontal angle. Assume that the total viewing angle interval is divided into n equal parts. If the difference between the camera poses projection horizontal angle and the first horizontal angle interval on the left side of the main direction is the smallest, it is considered that the camera’s horizontal angle is within the first interval at that time, which reduces the feature redundancy significantly. 

Moreover, five images from different angles are extracted equidistantly from a video stream image, as shown in Figure 6.

According to the above steps, the anchor feature database is constructed, and detailed information is provided in Table 2.

### 3.4. Feature Matching

To reduce the computation complexity of feature matching and matching errors, two categories are adapted: active anchor fast spatial indexing and feature screening.

#### 3.4.1. Active Anchors Fast Spatial Indexing

Currently, the popular feature indexing schemes include tree-based [38], graph-based [39], and hash-based schemes [40]. These schemes focus on the similarity of the feature descriptor rather than the spatial attribute in the feature point’s 3D environment. Torsten Sattler et al. [41] introduced the idea of active correspondence search, where, when the 2D–3D point matching is performed, the near points have a high matching probability. Inspired by this idea, this paper proposes a distributed spatial fast index algorithm that considers the additional anchor ID information and spatial coordinate information of the feature descriptor.

The parameters of the fast spatial indexing algorithm are shown in Algorithm 1. Assume that M feature points are extracted from a positioning image P, and m feature points are randomly selected from M feature points according to proportion k. Then, first, the K-D tree matching algorithm is used to match M feature points with the fingerprint database D, and the number of feature points that are successfully matched with the corresponding anchors and meet the matching threshold condition is calculated. The first n points with the largest counting number are selected as activation anchors. Then, only the features near the activation anchors are retained to obtain a reduced activation fingerprint database α. For the remaining (M − m) features, index matching is performed in the fingerprint database α, and the index matching results of all feature points are obtained by combining the former indexing results. If the statistical result of the second matching is significantly different from the first matching result, the feature extraction ratio k is decreased, and the above steps are repeated until the results of the two matchings are consistent.
**Algorithm 1: Fast Spatial Indexing****Input**: positioning image *P*, feature fingerprint database *D*, random *k*, threshold *r***Output**: best-matched anchor $C_{i}$ and $C_{i+1}$1: extract *M* features points from *P*2: **while** $C_{i}!=C_{i+1}$ **do**3:   decrease k4:   select m feature points from *M* according to rate *k*5:   match m features with *D* using the k-d tree algorithm6:   count the number of feature points from different anchors $N_{i}$ and record $C_{i}$7:    **if** $N_{i}$ > *r*8:      record the *ID* of this anchor and the corresponding $N_{i}$9:    **end if**10:     rank $N_{i}$, take the first *n*11:     extract anchors near *n* and narrow *D* into α12:     match *(m − n)* feature with *d*, record $C_{i+1}$13: **end while**

#### 3.4.2. Feature Screening

After the fast spatial indexing, approximate 3D matching points of the 2D points are obtained. However, the image features can change to a certain extent under the influence of illumination, distance, and angle of view, which can cause pseudo feature points of a 2D image and result in a mismatch. Therefore, feature screening is necessary before the pose estimation.

Since the homography matrix can be used to describe the affine transformation relationship between the positioning image and the reference image, the matching of the 2D coordinates of the same name points on the positioning and reference images can be realized, and the corresponding homography matrix can be calculated. If the homography matrix projection error is larger than threshold γ, then the matching is wrong; otherwise, the matching is correct.

According to the homography principle, it can be solved by at least four pairs of homonymous points. Let the coordinates of the kth pair of matching points in the positioning and reference images be $\left(u_{1}^{k},v_{1}^{k}\right)$ and $\left(u_{2}^{k}{,v}_{2}^{k}\right)$, respectively; then, the homography transformation is given by:(15)$$\left\{ \begin{array}{l} h_{1}u_{1}^{k}+h_{2}v_{1}^{k}+h_{3}-h_{7}u_{1}^{k}u_{2}^{k}-h_{8}v_{1}^{k}u_{2}^{k}=u_{2}^{k}\\ h_{4}u_{1}^{k}+h_{5}v_{1}^{k}+h_{6}-h_{7}u_{1}^{k}v_{2}^{k}-h_{8}v_{1}^{k}v_{2}^{k}=v_{2}^{k} \end{array}\right.$$

Further, the projection points of a position image $\left(u_{1},v_{1}\right)$ can be calculated using H and $\left(u_{1}^{\prime },v_{1}^{\prime }\right)$ from the reference image as follows:(16)$$\left[\begin{array}{c} u_{1}^{\prime }\\ v_{1}^{\prime }\\ 1 \end{array}\right]=\left[\begin{array}{ccc} h_{1} & h_{2} & h_{3}\\ h_{4} & h_{5} & h_{6}\\ h_{7} & h_{8} & 1 \end{array}\right]\left[\begin{array}{c} u_{1}^{}\\ v_{1}^{}\\ 1 \end{array}\right]$$

Therefore, the projection error ε of the above point can be calculated by:(17)$$\varepsilon ={\left(u_{1}^{\prime }-u_{1}\right)}^{2}+{\left(v_{1}^{\prime }-v_{1}\right)}^{2}$$

If ε is less than threshold γ, the obtained pair of matching points is considered correct. The results before and after feature screening are presented in Figure 7a,b, respectively. Finally, a user’s localization can be inferred accurately through the feature screening.

## 4. Experiment and Results

All experiments were performed at the THEOS remote sensing satellite ground station of the Silington Geospatial Information Science International Research Center of Wuhan University. The length of the floor plan was approximately 50 m, and the width was nearly 15 m. The hardware included a computer and a smartphone; the computer was equipped with a CPU with 16 GB memory and an Intel Core i7-4900 processor; the smartphone was a Huawei Mate20 with Kirin 980 CPU.

### 4.1. Experiment Setup

#### 4.1.1. Anchor Deployment Environment

A schematic diagram of the second floor of Silington ground station is presented in Figure 8, where the rectangle denotes the posted visual control anchor, and the total number of anchors is eight. The dot represents the indoor existing control point, which is the ground coordinates that are manually arranged by humans and measured by total station. The total number of indoor existing control points is 14. According to the distribution of indoor visual control anchor and control points, the path of the detailed surveys is planned, as shown in Figure 9. The red rectangle represents the visual control anchor, the blue rectangle denotes the visual measured anchor, and the orange straight line with the arrow indicates the detailed survey path. This path provides an order in which to measure the position and orientation information of visual anchors using ARscore.

#### 4.1.2. Localization Test Site

To test the positioning performance of the proposed algorithm, room 214/216 of Silington ground station was selected as an experiment site. Figure 10 shows the 3D schematic diagram of the selected room; the room was approximately 15 m long and 7 m wide, and there were 15 poster wallpapers in the room.

### 4.2. Result Analysis

#### 4.2.1. Performance Analysis of Feature Fingerprint Database

This study used the recall rate and the index time consumption of feature matching as evaluation indices of the feature fingerprint database (feature extraction algorithm) performance. The classic K-D tree feature matching algorithm was employed to perform a comparison experiment.

First, the parameters of the K-D tree were set as follows: the number of trees n was set to six, the type of evaluation distance was distType, and the nearest feature index result N was two. The images taken from different views were used as test data, and the fingerprint database (SURF features) with multi-view affine transform or without affine transform was used to index anchors. Because there could be features that did not belong to the anchor mural, such as the ceiling line and door frame line, which might reduce the recall rate, the positioning image was required to include only the anchor. 

The matching results of the above two groups of experiments are presented in Figure 11, where the x-axis represents the horizontal angle deviation when shooting, and the y-axis indicates the correct matching recall rate of an image feature.

As shown in Figure 11, the fingerprint database (SURF features) with affine transformation filtering performed better than the traditional SURF features in feature matching. When there is no angle deviation, the matching recall rates of the two methods were the same; when the change in the view angle reached 60°, the matching recall rate of the proposed database increased from 26% to 58%. The results indicate that the proposed database is robust and can improve the affine invariance significantly. In other words, the feature extraction algorithm with affine enhancement is more efficient than the traditional SURF.

#### 4.2.2. Anchor Fast Spatial Indexing Analysis

To compare the performance of the fast spatial indexing algorithm (the proposed algorithm) and the traditional K-D tree algorithm in feature matching, the indexing time and matching recall were used to quantify their performances. In this experiment, 300 images were collected at different control points and perspectives. 

In Figure 12, the blue and red curves represent the number of correctly matched feature points when the K-D tree algorithm and the spatial fast index algorithm were used, respectively. The red curve is higher than the blue curve, indicating that, compared with the K-D tree algorithm, the spatial fast index algorithm could significantly increase the number of correctly matched points.

The indexing time statistics of the two experiments are shown in Figure 13. Since the fast spatial indexing algorithm significantly compressed the fingerprint database, its indexing time was lower than that of the K-D tree algorithm. The average time consumption of the K-D-tree-algorithm-based indexing was 30.3 ms, while the average time of the fast spatial indexing algorithm was 12.7 ms, which is less than the half-time consumption of the K-D tree algorithm.

#### 4.2.3. Feature Screening Analysis

To select a suitable threshold to make the feature screening based on the homography matrix filtering result more efficient, images in places that were 1 m, 2–3 m, and more than 5 m away from the anchor point were selected, and the projection of matching points was performed. In addition, the projection error was calculated, as shown in Figure 14, where it can be seen that when the homography matrix filtering was performed, most of the matching points of the image taken 1 m away from the anchor had an error of 0–3 pixels, with an average error of 2.36 pixels. When the distance from the anchor was 2–3 m, the average error increased to 3.06 pixels; when the distance was more than 5 m, the average error increased to 3.51 pixels. Therefore, the error threshold was set to 5 pixels to reserve as many matching points as possible.

After determining the most suitable error threshold value, experiments were performed using the homography matrix screening algorithm to improve the correct matching rate. The inner points calculated by the later intersection solution were set as true values of the correct matching. The number of initial matching–homography screening point pairs was counted, as shown in Figure 15, where the statistical schematic diagram of the initial correct matching and the homography matrix screening point pair is presented.

As shown in Figure 15, after homography matrix filtering, the correct matching rate of most images reached 100%, and only a few images still had wrong matching points. According to the statistical results, the average initial correct matching rate values before and after the homography matrix screening were 87.47% and 99.8%, respectively, showing an increase of approximately 12%. Therefore, the homography matrix screening could significantly improve the feature matching performance.

#### 4.2.4. Localization Performance

To analyze the localization performance of the proposed method, eight control points in rooms 214/216 were selected and denoted by t1–t8. In addition, 326 images around these points with different view angles were selected to be used in the experiment. FILNet outputs about 254 localized images, which is about 78% of the total number of images. However, the image localization system using K-D tree as the matching method only outputs about 193 localized images, which is 59% of the total number of images. FILNet significantly improves the probability of successful localization compared to the K-D tree image localization method.

Figure 16 shows the ratio of localization images output by FILNet and baseline methods with different localization error ranges. From Figure 16, it can be seen that FILNet was higher than the baseline in the range of localization error below 0.1 m. Meanwhile, it was lower than baseline in the range of localization error above 0.1 m. FILNet and baseline output 94% and 83.9% of localization images with localization errors of less than 0.25 m, respectively. The localization accuracy was significantly improved.

To explore the influence of the distance from the anchor on the positioning accuracy when taking positioning images, collected positioning images at the distances of 0–2 m, 2–5 m, and more than 5 m were selected.

As shown in Figure 17, the errors were calculated. When the distance was 0–2 m, the positioning effect was the best, and the positioning error was less than 0.15 meters. When the distance was 2–5 m, the accuracy decreased slightly. When the distance was longer than 5 m, the proportion of the error of 0.3–1 m increased obviously, but the overall positioning accuracy was acceptable. It can be seen from the above table that the anchor-based positioning algorithm had a good performance on the whole. Among them, 95% of the image positioning errors were within 0.3 m, and only 1.08% of the positioning errors exceeded 1 m. As for the average error, 96.46% of the errors were less than 0.3 m. Therefore, the method can enhance localization performance effectively.

### 4.3. Discussion of Limitations

Although our proposed FILNet algorithm showed a significant decrease in feature matching errors and localization errors, several issues remain to be discussed: (1) Lighting conditions have a significant impact on FILNet localization systems. Low, uneven, or changing lighting can reduce the number of features detected, degrade feature quality, and increase mismatches. (2) Camera motion blur caused by shaking or fast movement can distort image features and reduce localization performance. (3) The distance between the online query image and the closest position in the image database also affects localization accuracy. As the distance increases, the perspective difference causes greater feature distortion and mismatch. Beyond a certain distance threshold, the similarity in feature appearance diminishes, and correct matches become difficult.

## 5. Conclusions and Future Work

### 5.1. Conclusions

In this article, we proposed a fast image-based indoor localization approach using the anchor control network, which combines an affine invariance enhancement algorithm and spatial fast indexing to improve robustness and efficiency. First, murals are deployed on the wall as the anchors; some of them are selected as the control anchors, which are measured by the total station, and the rest of the anchors are inferred by visual–inertial odometry based on ARcore. Then, we innovatively introduced the affine invariance enhancement algorithm to supply multi-angle features for a single anchor, which made the system more robust and accurate in feature matching. Furthermore, to overcome the complexity of feature matching, we developed a spatial fast indexing method to lower the computation complexity. Finally, our approach was evaluated in real-world tests; the results demonstrated higher accuracy, less time consumption, and better robustness in response to the view angle changes.

### 5.2. Future Work

This paper presented a set of indoor positioning schemes based on plane anchors where the offline database building is used for online positioning, realizing high-precision positioning. However, there are still certain deficiencies that need to be improved in the future. First, in the offline stage, the visual–inertial odometer is used to measure anchors, which relies heavily on indoor light. Understanding how to ensure the effectiveness of database construction when the indoor light is insufficient could be the focus of future research. Second, during fast spatial indexing, the size of the fingerprint database is likely to reduce sharply after the per-step screening, which can further reduce the index efficiency of the K-D tree algorithm. Third, when anchor points are deployed at other locations, they may be affected by external factors. For example, anchor points cannot be deployed or are deployed in insufficient numbers, which may lead to localization failure. At this time, it can try to use the shot image of the scene instead of the anchor image for building the fingerprint library to complete the localization. Fourth, in the case where the anchor fingerprint library for the scene has already been deployed, some anchors may have been obscured, moved, or removed. It is not necessary to reconstruct the fingerprint library, and the method in this paper can still be applied. We only need to update the new coordinates of the moving anchors or add new anchors to the fingerprint library to complete the localization system.

In the future, we will expand the types of visual anchors and will consider some objects with salient features that already exist in the indoor environment to ensure that each positioning image contains at least one visual anchor. Meanwhile, we will also explore multi-modal sensors fusion to perceive other anchors and make the system more feasible. Finally, how the distance between anchors or the density of anchors affects the accuracy of localization will be investigated as well in our future work.

## Figures and Tables

**Figure 1 sensors-23-08140-f001:**
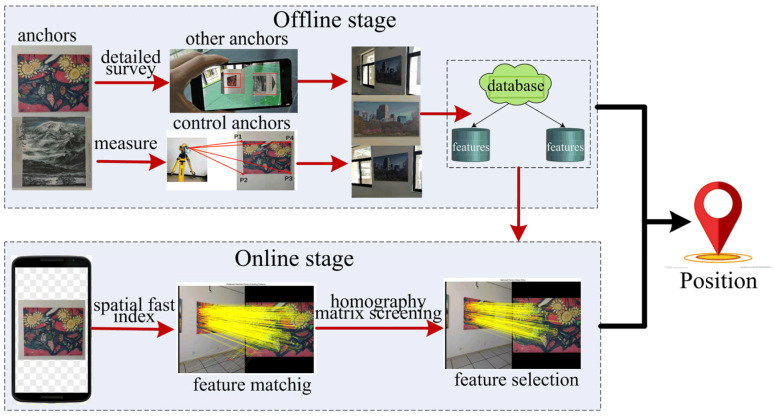
The overview of the proposed system.

**Figure 2 sensors-23-08140-f002:**
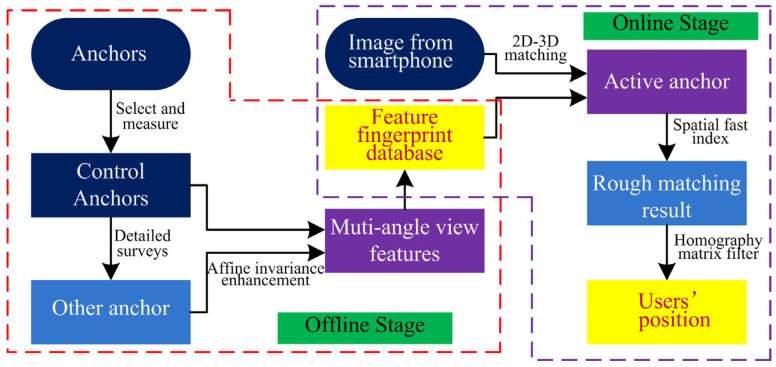
The framework of the proposed method.

**Figure 3 sensors-23-08140-f003:**
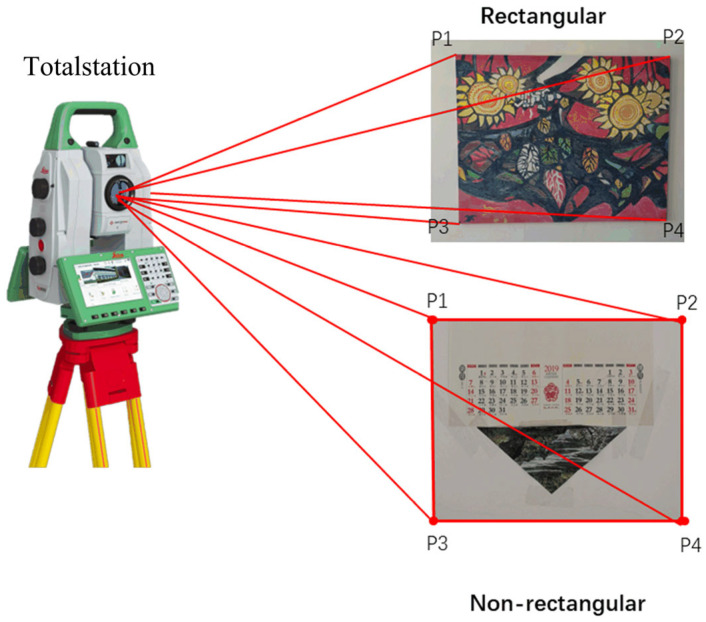
The control anchor measurement.

**Figure 4 sensors-23-08140-f004:**
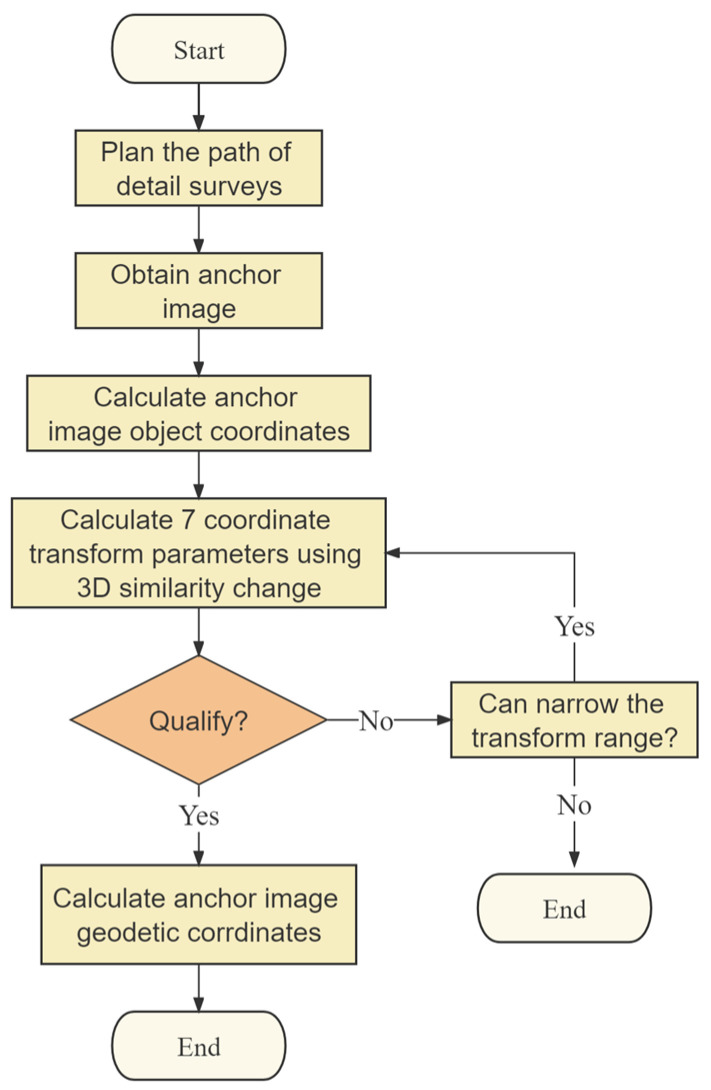
The flowchart of detailed surveys.

**Figure 5 sensors-23-08140-f005:**
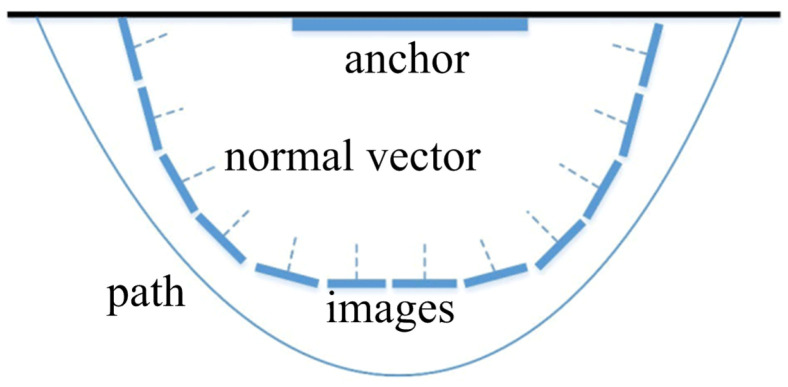
The image collecting path process illustration.

**Figure 6 sensors-23-08140-f006:**
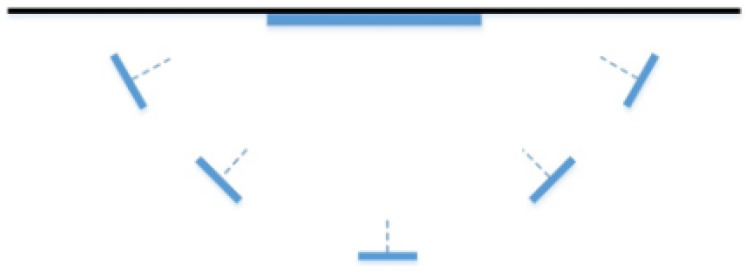
The result at *n* = 5.

**Figure 7 sensors-23-08140-f007:**
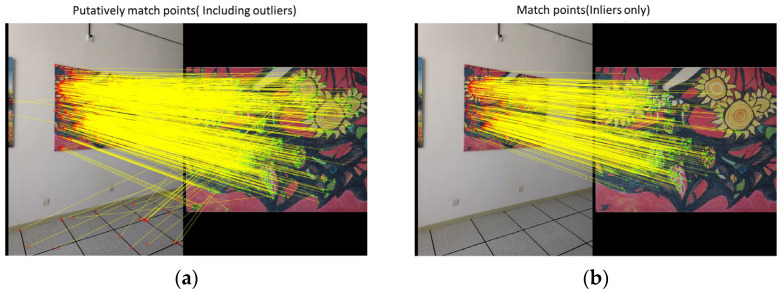
Feature matching results. (**a**) Before screening. (**b**) After screening.

**Figure 8 sensors-23-08140-f008:**
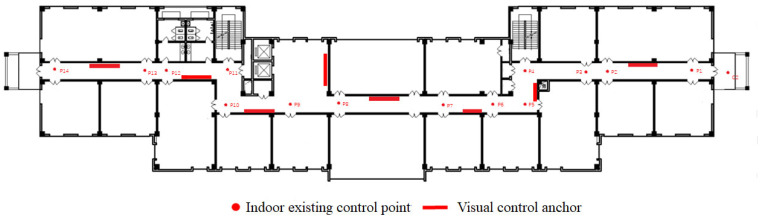
The floorplan with anchors. P1–P14 represent the existing control points, and d2 is a symbol of the outside.

**Figure 9 sensors-23-08140-f009:**
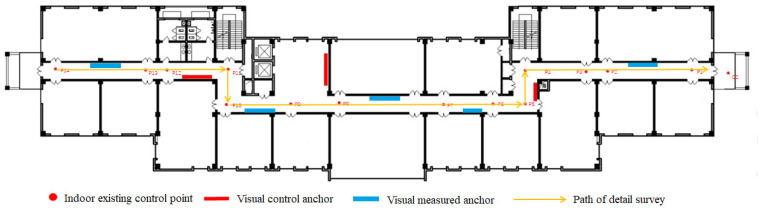
The floorplan with anchors and detailed survey path. P1–P14 represents the existing control points and d2 is a symbol of the outside.

**Figure 10 sensors-23-08140-f010:**
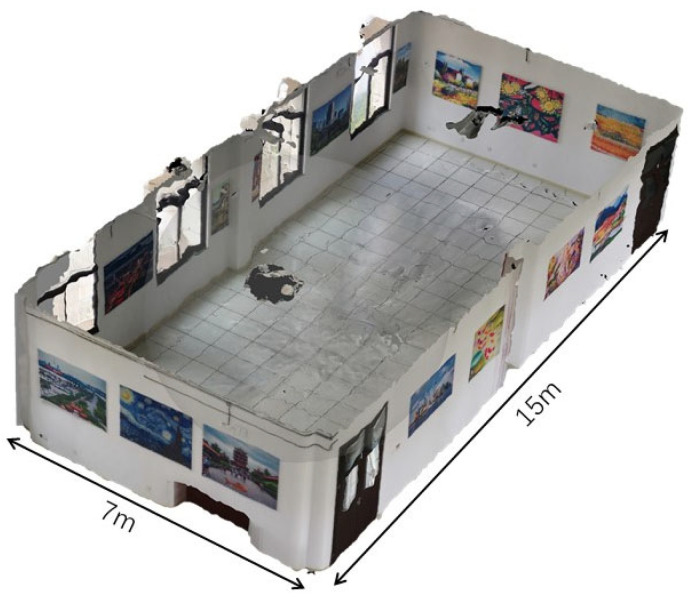
The 3D illustration of room 214/216 in Silington ground station.

**Figure 11 sensors-23-08140-f011:**
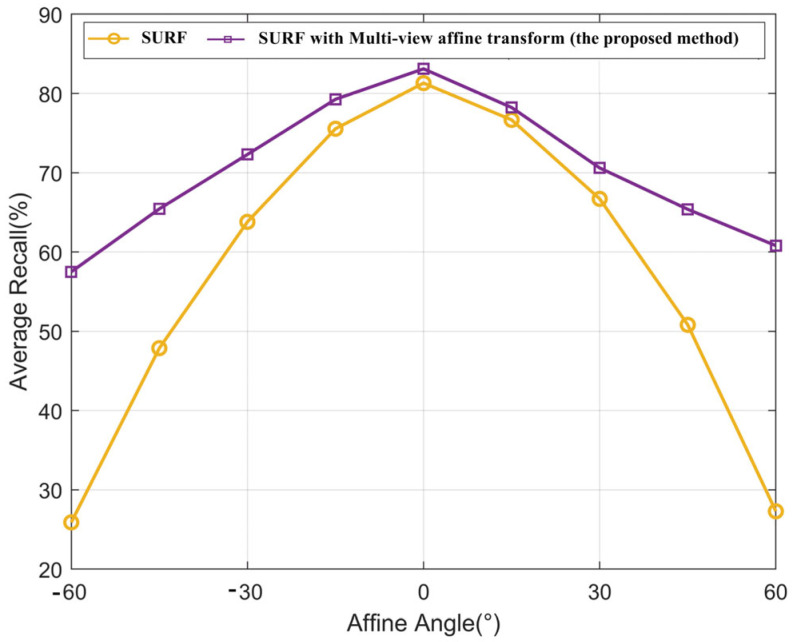
Recall changes with view angle.

**Figure 12 sensors-23-08140-f012:**
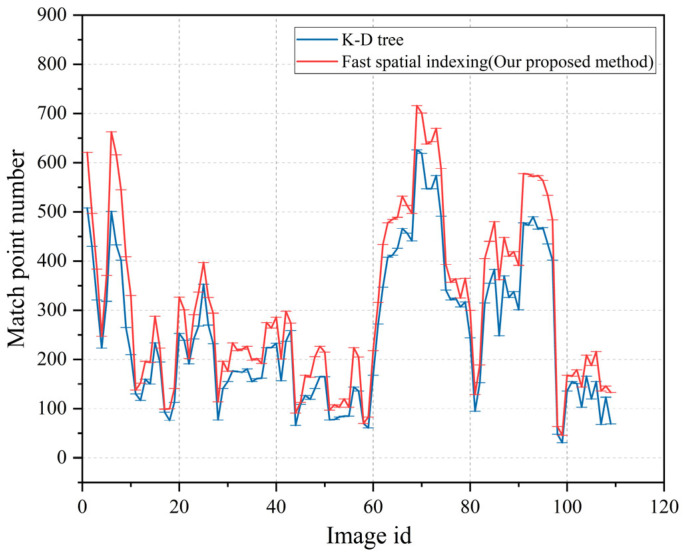
The number of matching points from different images.

**Figure 13 sensors-23-08140-f013:**
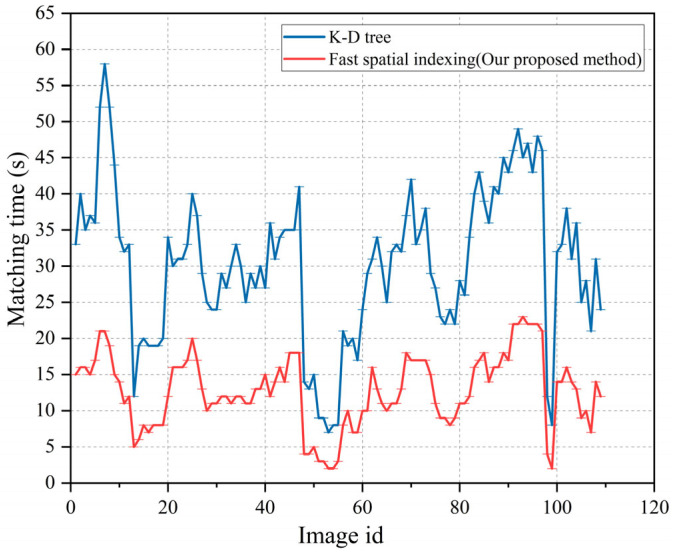
The time consumption results of the two algorithms.

**Figure 14 sensors-23-08140-f014:**
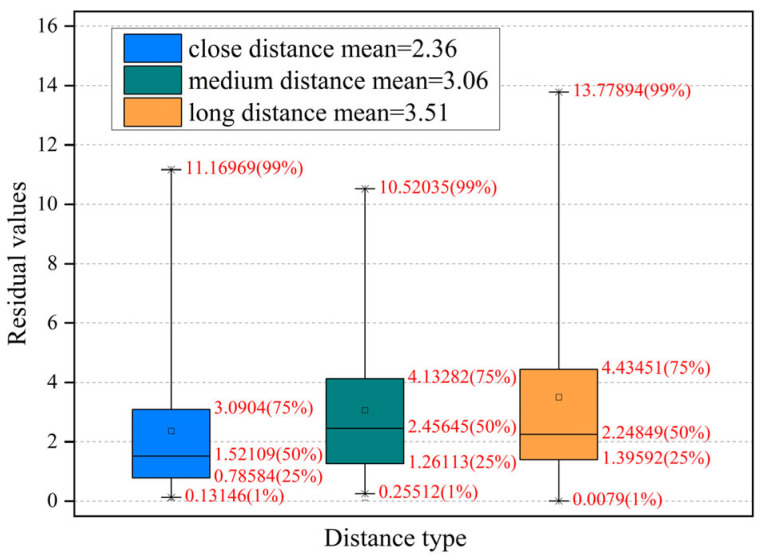
Residual value changes with distance.

**Figure 15 sensors-23-08140-f015:**
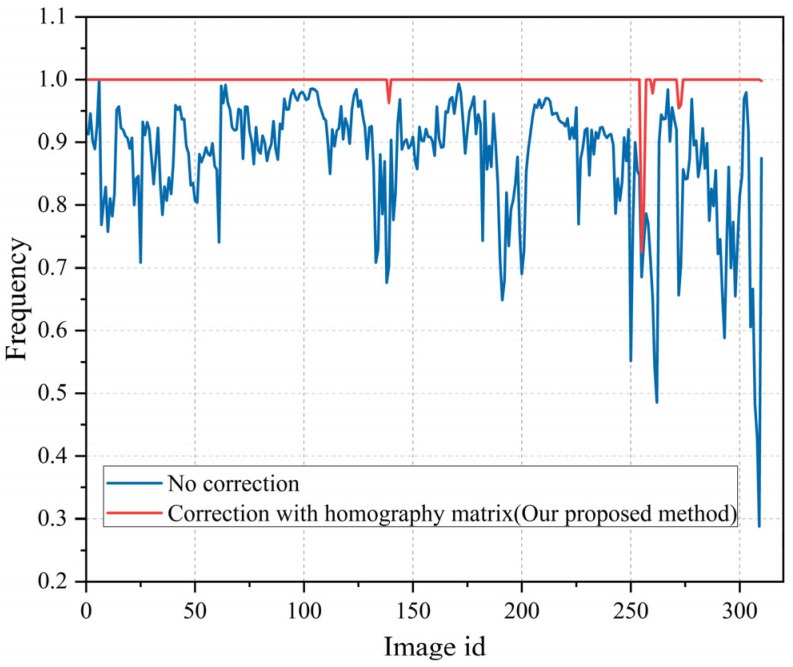
Matching rate of the two algorithms.

**Figure 16 sensors-23-08140-f016:**
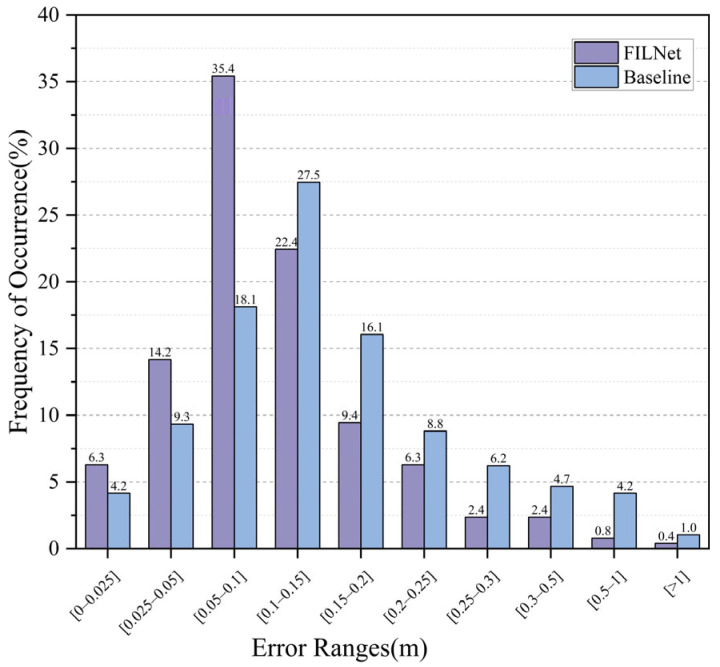
Localization error with the view angle changes.

**Figure 17 sensors-23-08140-f017:**
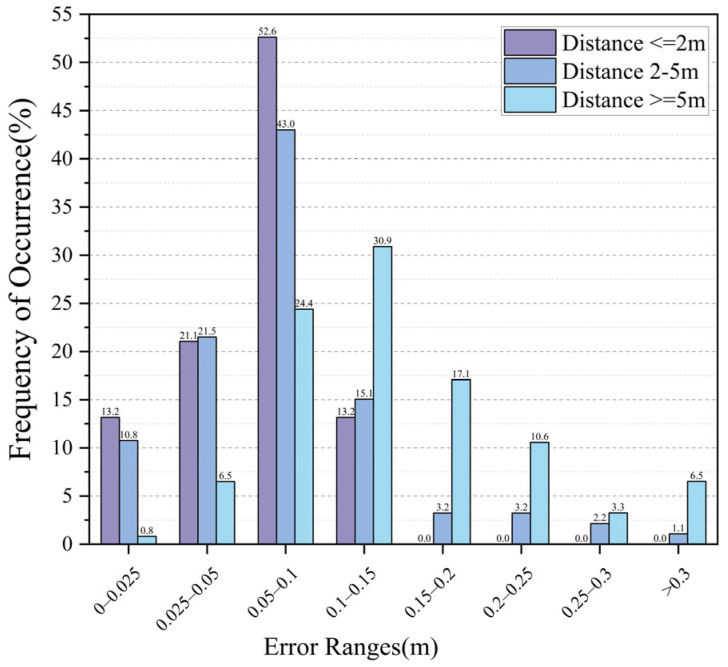
Localization error at different distances.

**Table 1 sensors-23-08140-t001:** Performance comparison between the three features extraction algorithms.

Algorithm	Speed	Robustness		
		Rotation	Angle	Scale
SIFT	low	better	better	better
SURF	faster	best	best	best
ORB	fastest	better	better	/

**Table 2 sensors-23-08140-t002:** Detailed information on the feature fingerprint dataset.

Parameter	Value
Feature descriptor	64 dimensions
2D plane coordinate	a (u, v)
3D object coordinate	A (x, y, z)
Anchor ID	001

## Data Availability

Not applicable.

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
