# Peer review of "FILNet: Fast Image-Based Indoor Localization Using an Anchor Control Network"

_sensors, 2023, doi:10.3390/s23198140_

Round 1

Reviewer 1 Report

Reviewer report

This paper designs a fast image-based indoor localization method based on an anchor control network to improve localization accuracy and shorten the time of feature matching. The contribution of the paper, if any, has not been significant. The paper is well-written and organized. To enhance the paper's contribution, addressing the following suggestions is recommended:

11. Emphasize the main focus of the current study, which is the localization error, by incorporating significant results related to localization error into the abstract.

  2. Clearly highlight in the introduction section the gap left by previous works and how the current work addresses this gap.

33. Include a discussion about the ZigBee wireless protocol in the introduction section, as it holds significance for indoor localization. Relevant insights can be found at: https://doi.org/10.1016/j.measurement.2020.108276.

44. Explore the impact of light intensity on localization and incorporate this discussion.

55. Revise the flow chart depicted in Figure 4 to ensure the inclusion of start and stop blocks.

66. Provide definitions for equation variables before or after each equation and reorganize equation numbers on the right side of the paper.

77. Ensure clear presentation of points in Figures 9 and 8.

88.  Incorporate the dimensions of room 214/216 into Figure 10.

99.  Discuss why there is a higher frequency occurrence of localization error in the range of 0.05 - 0.1 compared to other ranges. Furthermore, it is advisable to display the figure in the form of a histogram.

110. Enhance the text resolution in the result figures section.

111. Correct the repetition of the title heading "5. Experiment and Results" in both Section 4 and Section 5 at line 550.

112. Update outdated references, such as refs 8, 10, 14, 34, 35, and 36.

None

Author Response

Dear Reviewers:

We would first like to thank you for your helpful comments and suggestions. To
improve the quality of the manuscript, we have taken into account all of the comments raised by the reviewers, and we have thoroughly revised our paper. We hope that it has now reached the high standard of the famous journal. Please see our detailed point-bypoint replies below (in blue). All the modifications in the revision are highlighted in blue to help the editors and reviewers in finding the changes with regard to the previous version.

Thanks for your time.

Sikang Liu, Zhao Huang, Jiafeng Li, Anna Li, and Xingru Huang.

Reviewer 2 Report

Comments to the Author

This paper proposes a fast image-based indoor localisation method based on an anchor control network (FILNet) to improve localisation accuracy and shorten the time of feature matching. It is an interesting research topic with many potential application areas. However, there are several points that need to be addressed to improve the quality of the manuscript.

Suggestions to improve the quality of the paper are provided below:

1.     It was mentioned very briefly in the first paragraph of the Introduction section that indoor location-based services have been extensively studied due to their large application prospect. However, the authors did not go into any details about any these application areas. In order to make this paper more relatable to a wider audience of readers, the authors should briefly describe some of these application areas, which include emergency management, smart energy management, smart HVAC controls and occupancy detection. I suggest that the authors review the following established works as a good starting point to highlight the important application areas where indoor positioning systems are leveraged.

Indoor localisation for building emergency management

10.1109/IUCC-CSS.2016.013

Indoor localisation for smart energy management

https://doi.org/10.1016/j.buildenv.2022.109472

Indoor localisation for smart HVAC controls

https://doi.org/10.1145/2517351.2517370

Indoor localisation for point-of-interest identification

https://doi.org/10.3390/ijgi10110779

Indoor localisation for occupancy prediction

https://doi.org/10.1016/j.buildenv.2022.109689

2.     While the authors have attempted to compare the various indoor localisation technologies in the Introduction section, it appears to be lacking as it only mentions about the limitations of other approaches while only highlighting the advantages of image-based localisation approaches. On the other hand, it was highlighted by [1] that image-based approaches are limited due to privacy concerns and it also requires a clear line of sight with minimum obstructions to achieve good performance. Furthermore, this technology also requires the deployment of advanced signal processing and expensive hardware. Please kindly review the following paper as a starting point and explain how these limitations will be addressed in this paper.

[1] https://doi.org/10.1016/j.buildenv.2020.106681

3.     Aside from simply reviewing past works in the Related Work section, I suggest that the authors put in more thought into the limitations of these works and discuss how they will addressed in this manuscript.

4.     Why does the K-D tree feature matching algorithm serve as a good baseline for model comparison?

5.     In Figure 13, is a particular reason why there are larger differences in matching times for certain images between K-D tree and the proposed fast spatial indexing approach, while some images have smaller differences? Furthermore, the matching time for the K-D tree method seem to vary significantly between 10-60 (ms?) while the proposed method seem to be more robust. Why is that the case?

6.     Why are the results for the baseline approach (i.e., K-D tree algorithm) not reported in Section 4.2.3 and 4.2.4?

7.     In the last section on Future Work, please discuss about the following limitations as well. For instance, given that one of the key task in the proposed indoor localisation system involves constructing an anchor fingerprint database, how does this requirement affect the scalability of the approach when deploying to other locations? What happens if some of these anchors are obstructed, shifted, removed? Does the database need to be reconstructed or is the current approach robust enough for such scenarios?

8.     Minor comments

·       Increase the figure sizes in the Results section.

·       The section name for Section 5 “Experiment and Results” is a repeat of Section 4. Please change it to something more relevant to Conclusion and Future Work.

There are no major issues related to the manuscript's quality of English, except for some minor issues highlighted in my current set of comments.

Author Response

(The authors gave the same response as above.)

Round 2

Reviewer 1 Report

The paper has been revised by the authors in accordance with my comments. Subsequently, the list of references should be revised to align with the updated version. Once the references have been updated, the paper will be ready for acceptance for publication.

Author Response

We thank the reviewers for their positive comments. In the revised manuscript, we have seriously addressed the issue of reference updating raised by the reviewers. We have made great efforts to improve our work. The responses to the questions are presented as follows.

Reviewer 2 Report

Thank you for taking the time to address my comments thoroughly and comprehensively. I believe all my comments have been adequately addressed, and the quality of the manuscript has increased significantly as a result. I have determined that the manuscript is now ready for publication.

There are no major issues related to the manuscript's quality of English, except for some minor issues that do not affect the clarity and flow of the manuscript.

Author Response

(The authors gave the same response as above.)
